# Quantum spin Hall phase in 2D trigonal lattice

Z.F. Wang[1,2], Kyung-Hwan Jin[2] & Feng Liu[2,3]

The quantum spin Hall (QSH) phase is an exotic phenomena in condensed-matter physics. Here we show that a minimal basis of three orbitals ($s$, $p_x$, $p_y$) is required to produce a QSH phase via nearest-neighbour hopping in a two-dimensional trigonal lattice. Tight-binding model analyses and calculations show that the QSH phase arises from a spin–orbit coupling (SOC)-induced $s$–$p$ band inversion or $p$–$p$ bandgap opening at Brillouin zone centre ($\Gamma$ point), whose topological phase diagram is mapped out in the parameter space of orbital energy and SOC. Remarkably, based on first-principles calculations, this exact model of QSH phase is shown to be realizable in an experimental system of Au/GaAs(111) surface with an SOC gap of $\sim 73$ meV, facilitating the possible room-temperature measurement. Our results will extend the search for substrate supported QSH materials to new lattice and orbital types.

[1] Hefei National Laboratory for Physical Sciences at the Microscale, Synergetic Innovation Center of Quantum Information and Quantum Physics, University of Science and Technology of China, Hefei, Anhui 230026, China. [2] Department of Materials Science and Engineering, University of Utah, Salt Lake City, Utah 84112, USA. [3] Collaborative Innovation Center of Quantum Matter, Beijing 100084, China. Correspondence and requests for materials should be addressed to Z.F.W. (email: zfwang15@ustc.edu.cn) or to F.L. (email: fliu@eng.utah.edu).

Currently, there are two prevailing theoretical models for the quantum spin Hall (QSH) phase in a two-dimensional (2D) system: the Kane–Mele model[1] and the Bernevig–Hughes–Zhang (BHZ) model[2,3]. In Kane–Mele model, the QSH phase is realized by any finite SOC-induced bandgap opening at Dirac point, as a generalization of Haldane's model[4] to spinful system with time reversal symmetry in a hexagonal lattice. In BHZ model, the QSH phase is realized by SOC-induced band inversion at time reversal invariant momenta between two bands of different parities, originally derived from a square lattice of HgTe quantum-well system. However, 10 years after these original theoretical proposals, only two material systems with a tiny bandgap (HgTe/CdTe (ref. 3) and InAs/GaSb (ref. 5)) have been confirmed experimentally for the BHZ quantum-well model, while no experiment has yet to confirm the Kane–Mele model in a real material of hexagonal lattice, despite many material systems have since been theoretically predicted[6–23]. Therefore, there remains an intensive search for QSH materials, especially with a large gap, and it is highly desirable to expand such search beyond the original hexagonal and square lattice to increase the feasibility for experimental realization.

Here we prescribe a discrete lattice model for QSH phase in a trigonal lattice. We consider a minimal basis of three orbitals ($s$, $p_x$, $p_y$) per lattice site of trigonal symmetry with nearest-neighbour hopping, solve an effective tight-binding Hamiltonian and develop a generic phase diagram for the non-trivial band topology in the parameter space of orbital energy and SOC. Depending on the order of $s$ versus $p$ orbital energies, a QSH phase may arise from either strong or weak SOC. Most remarkably, based on first-principles calculations, this exact QSH model is shown to be possibly realizable in an experimental system of Au/GaAs(111)[24] with a large non-trivial SOC gap of $\sim 73$ meV.

## Results

**The minimal basis QSH lattice model.** In general, a minimal basis of three orbitals ($s$, $p_x$, $p_y$) in a trigonal lattice is shown in Fig. 1a, which can also be equivalently transformed into three $sp^2$ hybridized orbitals, as shown in Fig. 1b. The corresponding tight-binding Hamiltonian in the basis of ($s$, $p_x + ip_y$, $p_x - ip_y$) is shown in Supplementary Note 1. Around the $\Gamma$ point, expending to the first-order of $k$ the Hamiltonian for spin-up band reduces to

$$H = \begin{pmatrix} \varepsilon_s + 6t_{ss\sigma} & \frac{3}{\sqrt{2}}t_{sp\sigma}(ik_x - k_y) & \frac{3}{\sqrt{2}}t_{sp\sigma}(ik_x + k_y) \\ \frac{3}{\sqrt{2}}t_{sp\sigma}(-ik_x - k_y) & \varepsilon_p + 3(t_{pp\sigma} + t_{pp\pi}) + \lambda & 0 \\ \frac{3}{\sqrt{2}}t_{sp\sigma}(-ik_x + k_y) & 0 & \varepsilon_p + 3(t_{pp\sigma} + t_{pp\pi}) - \lambda \end{pmatrix}$$

(1)

where $\varepsilon_s$ and $\varepsilon_p$ are on-site energies for $s$ and $p$ orbitals, respectively. $t_{ss\sigma}$, $t_{sp\sigma}$, $t_{pp\sigma}$ and $t_{pp\pi}$ are nearest-neighbour hopping parameters and $\lambda$ is SOC strength. $k_x$ and $k_y$ are momentum along $x$ and $y$ direction. Solving equation (1), the three eigenvalues are $E_s = \varepsilon_s + 6t_{ss\sigma}$ and $E_p^{\pm\lambda} = E_p \pm \lambda = \varepsilon_p + 3(t_{pp\sigma} + t_{pp\pi}) \pm \lambda$. Clearly, the $s$ orbital is independent of SOC and the two $p$ orbitals are degenerate without SOC. Depending on the order of $E_s$ versus $E_p$, there are two different types of bands.

For the first type, the $s$-band is above the $p$-band ($E_s > E_p$) without SOC, as shown in Fig. 1c. The red and blue colours indicate the components of $s$ and $p$ orbitals, respectively, and the parities for each sub-band at time reversal invariant momenta[25] ($\Gamma$ point and three M points) are labelled with $+$ and $-$ signs. It is found that the topological invariant[25] is $Z_2 = 0$ for middle and bottom two sub-bands, indicating clearly a normal insulator phase. To realize the QSH phase, an SOC-induced $s$–$p$ band inversion is needed ($E_s < E_p^\lambda$), that is, first closing of a trivial gap followed by reopening of a non-trivial gap by including the SOC.

With the increasing SOC strength, one can see that the bandgap between top and middle sub-band first reduces and closes (Fig. 1d), and then reopens (Fig. 1e) at the $\Gamma$ point. The $s$–$p$ band inversion induces a parity exchange in this process, so that the topological invariant changes from $Z_2 = 0$ (Fig. 1c) to $Z_2 = 1$ (Fig. 1e) for the middle and bottom two sub-bands.

For the second type, the $s$-band is below the $p$-band ($E_s < E_p$) without SOC, as shown in Fig. 1f. Different from Fig. 1c, the topological invariant is already $Z_2 = 1$ for the middle and bottom two sub-bands, that is, the band order has already been inverted even without SOC. Thus, any finite SOC will open a non-trivial gap at Dirac point to turn the system into a QSH phase as the case for graphene[1], as shown in Fig. 1g. This second type of QSH phase in a trigonal lattice has also been recently discussed with multiple $p$-bands in a $k \cdot p$ model[26]. The parity does not change by including the SOC in this process. On the basis of the above band order analysis, a topological phase diagram can be constructed, as shown in Fig. 1h. Using $\Delta = E_s - E_p$ and $\lambda$ as two independent parameters, normal insulator and QSH phase is divided by the bandgap closing line ($E_s = E_p^{+\lambda}$, dashed line in Fig. 1h). The parameters used for band structure in Fig. 1c–g are labelled with I–V. We have marked these data points in the phase diagram (Fig. 1h), with dashed arrows indicating the increasing SOC strength to distinguish these two different QSH phase realization processes. Thus, we introduce a discrete lattice model with minimal basis of three orbitals in a trigonal lattice to realize the QSH phase.

**The effective QSH model.** The band structure evolution in Fig. 1c–g (I–V) can be seen more intuitively from a three-dimensional (3D) band plot around the $\Gamma$ point, as shown in Fig. 2a,d for the first and second type of band order, respectively. It has been shown that different lattices models may be adiabatically connected to different classes of effective models of topological quantum field theory at the continuum limit, which generally requires only two-band inversion[27]. Correspondingly, one can reduce the three-band Hamiltonian to an effective two-band Hamiltonian around the $\Gamma$ point[27] as $H_{\text{eff}} = d_0(I) + \mathbf{d} \cdot \boldsymbol{\sigma}$, where $I$ is the identity matrix and $\boldsymbol{\sigma}$ is the Pauli matrices, $d_0$ is a parameter and $\mathbf{d}$ is a 3D vector field in momentum space (Supplementary Note 2). For the non-trivial band III (Fig. 2a) and V (Fig. 2d), $\hat{\mathbf{d}} = \mathbf{d}/|\mathbf{d}|$ has a vortex structure around the $\Gamma$ point, as shown in Fig. 2b,e, respectively. At the $\Gamma$ point, $\hat{\mathbf{d}}$ is along south pole in Fig. 2b and north pole in Fig. 2e. When $k$ goes away from the $\Gamma$ point, $\hat{\mathbf{d}}$ changes direction gradually from out of plane to in-plane in both cases. In addition, the energy splitting between the two bands is characterized by the value of $2|\mathbf{d}|$, which is $|E_s - E_p^{+\lambda}|$ and $2\lambda$ at the $\Gamma$ point for band III and V, respectively. Generally, a vortex is a topological defect described by the Chern number ($C$). To confirm this, we also calculate the Berry curvature for band III and V, as shown in Fig. 2c,f, respectively. The Berry curvature is non-zero around the $\Gamma$ point and gives $C = 1$, confirming a non-trivial topological phase.

**The first-principles calculations.** After establishing the above QSH model, the next question is whether it can be realized in a real material? Instead of searching for a free-standing 2D material, which is generally metastable and whose intrinsic topological properties can be altered when it is placed on a substrate, we opt to focus on surface-based 2D materials that is supported on a substrate, that is, it is atomically bonded to but electronically decoupled from the substrate[15–19,28,29], so that it is relatively more feasible for experimental realization, as well as

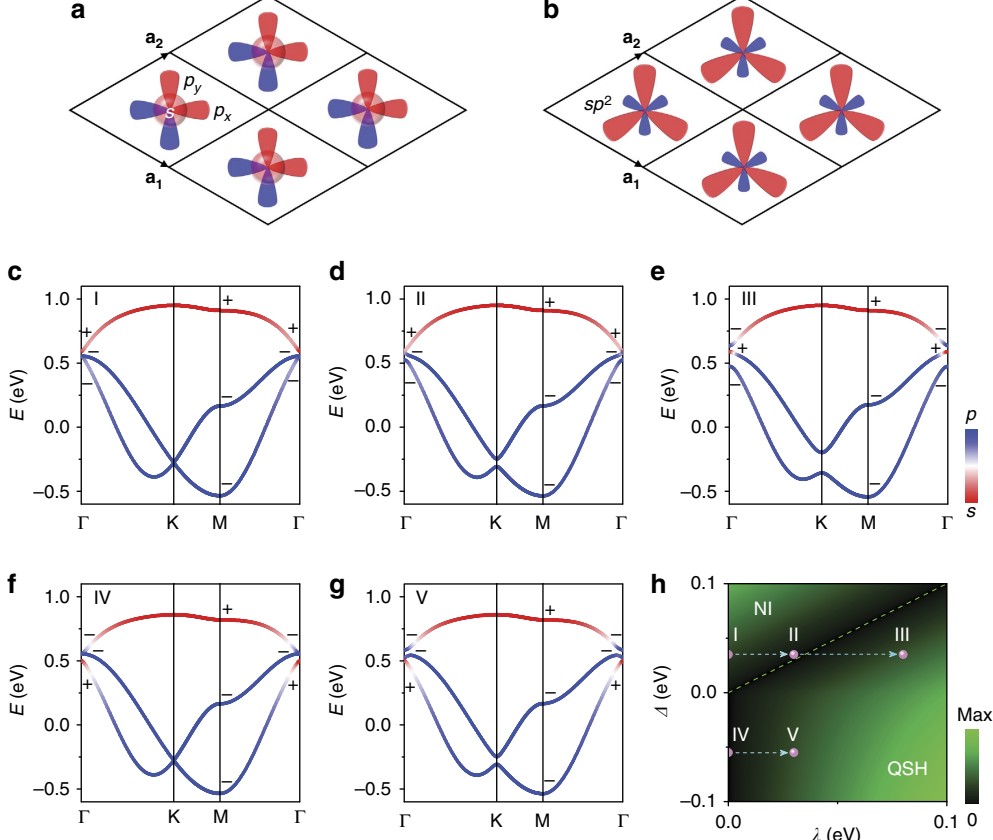

**Figure 1 | Minimal basis tight-binding model for QSH phase in a trigonal lattice.** (**a,b**) Trigonal lattice with three orbitals ($s$, $p_x$, $p_y$) per lattice site and its equivalent three $sp^2$ orbitals. $\mathbf{a_1} = (\sqrt{3}/2, -1/2)$ and $\mathbf{a_2} = (\sqrt{3}/2, 1/2)$ are lattice vectors. (**c-e**) The first-type band structures with parameter $\varepsilon_s = 0.83$ eV, $\varepsilon_p = 0$ eV, $t_{ss\sigma} = -0.04$ eV, $t_{sp\sigma} = 0.09$ eV, $t_{pp\sigma} = 0.18$ eV and $t_{pp\pi} = 0.005$ eV. $\lambda$ is 0, 0.03 and 0.08 eV for **a,b** and **c**, respectively. (**f,g**) The second-type band structures with $\varepsilon_s = 0.74$ eV and $\lambda = 0$, 0.03 eV for **f,g** respectively. The other parameters are the same to those in **c-e**. From **c-g**, the red and blue colours indicate the component of $s$ and $p$ orbitals, respectively, and the parities for each sub-band at time reversal invariant momenta are labelled with + and − signs. (**h**) Topological phase diagram in the parameter space of $\Delta = E_s - E_p$ (bandgap between $s$ and $p$ orbitals at $\Gamma$ point without SOC) and $\lambda$ (SOC strength). The colour indicates the bandgap between top and middle band. The band structure parameters for **c-g** are marked by the dots with labels I–V in **h**. The dashed line is the boundary between normal insulator (NI) and QSH phase. The dashed arrows indicate the increasing SOC strength.

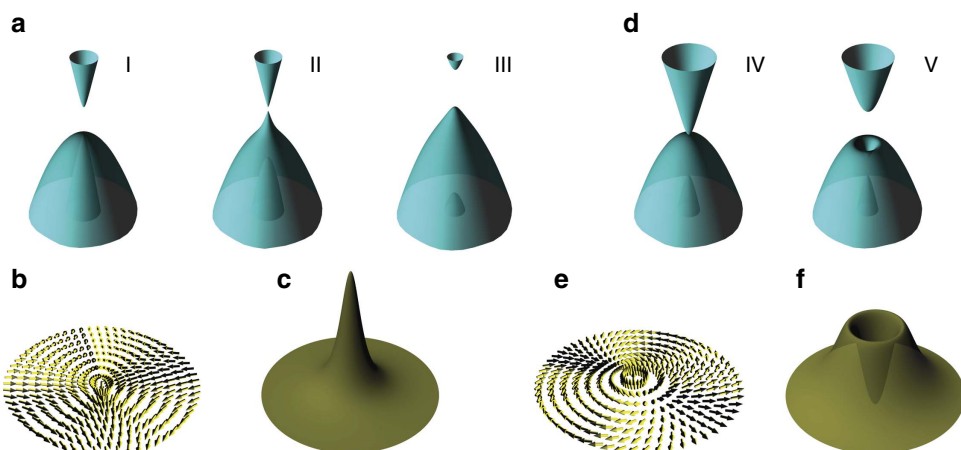

**Figure 2 | Band, vortex and Berry curvature around the Γ point.** (**a**) 3D band structure around the Γ point for bands I-III, illustrating the $s$–$p$ band inversion process. (**b,c**) Vortex structure of $\hat{\mathbf{d}}$ and Berry curvature around the Γ point for the effective ($s$, $p$) two-band model of band III in **a**. (**d**) 3D band structure around the Γ point for bands IV and V, illustrating the $p$–$p$ bandgap-opening process. (**e,f**) Vortex structure of $\hat{\mathbf{d}}$ and Berry curvature around the Γ point for the effective ($p$, $p$) two-band model of band V in **d**.

easier to facilitate experimental measurement and device fabrication. Also, we note that usually metal atoms favour a close-packing geometry forming trigonal lattice in 2D rather than the open hexagonal lattice. In fact, many heavy metal atoms grown on a semiconductor surface have been found to exhibit a trigonal lattice symmetry, such as Bi and Tl on Si(111)[30], Au on Ge(111)[31].

Specifically, the system of Au/GaAs(111) has drawn our attention. Hilner et al.[24] showed that at low coverage of Au on the As-terminated GaAs(111) surface, a $\sqrt{3}\times\sqrt{3}R30°$ trigonal superlattice structure was observed by scanning tunnelling microscopy. Furthermore, a theoretical model with one Au atom adsorbed on every third hexagonal close-packed threefold hollow site of the Ga lattice is shown to be the most energetically stable, with a simulated scanning tunnelling microscopy image in excellent agreement with the experiment (Fig. 1c in ref. 24). So, starting from the known structural model of Au/GaAs(111)[24] (Fig. 3a), its topological properties are systematically studied. The band structure of Au/GaAs(111) without SOC is shown in Fig. 3b. The key feature of notice is the Dirac band around the $\Gamma$ point above the Fermi level (a 3D band plotting is shown in the inset of Fig. 3b), which are well separated from the other bulk bands (shaded region). The $s$ and $p$ orbital components for band I, II and III (labelled in Fig. 3b) are shown by red and blue colours, respectively. At the $\Gamma$ point, band I, II and III are mainly made of $p$, $p$ and $s$ orbitals. When $k$ goes away from the $\Gamma$ point, band I, II and III are mainly made of $s$, $p$ and $p$ orbitals. The overall band shape and orbital components are found to be consistent with the band IV (Fig. 1f), as obtained from the minimal basis model in a trigonal lattice discussed above. This is further confirmed by plotting the real-space charge density distribution of these three bands at the $\Gamma$ point, as shown in Fig. 3c. For band I and II, the charge densities are mainly localized on top of three As atoms, showing an in-plane $p$ orbital shape. For band III, the charge densities are mainly localized on Au atom, showing an $s$ orbital shape. So, effectively these three bands arise from three orbitals of ($s$, $p_x$, $p_y$) character (Fig. 1b) with nearest-neighbour hopping in a trigonal lattice.

Quantitatively, to obtain a better fitting for the first-principles bands (band I, II and III) of Au/GaAs(111), a $3 \times 3$ Wannier Hamiltonian is constructed by using the maximally localized Wannier functions (MLWFs) in Wannier90 package[32]. As shown in Fig. 3d, the Wannier band shows excellent agreement with the first-principles band. The three fitted MLWFs are shown in Fig. 3e, which are equivalent to each other and have a threefold rotational symmetry around centre Au atom. Each MLWF has mixed components from both As-$p$ and Au-$s$ orbitals. Adding the components together, the overall orbital shape is shown in the last panel of Fig. 3e. An $s$-type orbital is centred at Au atom, and three tilted $p$-type orbitals are centred at three surface As atoms along the As–Au bond direction, forming effectively an $sp^2$ hybridization. This can be easily understood from the un-hybridized surface orbitals. As schematically shown in Fig. 3f, each Au atom contributes one $s$ orbital, each dangling bond of surface As atom contributes one $sp^3$ orbital. The $sp^3$ orbital can be further decoupled into the orbitals along and perpendicular to the Au–As bond directions. Because of the odd orbital symmetry, the perpendicular component has a negligible hopping integral with Au-$s$ orbital, while the other component along the bond has a large overlap with the Au-$s$ orbital to form a $\sigma$ bond.

**The analysis of non-trivial topological phase.** Generally, when SOC is introduced to the Dirac band, a non-trivial topological phase is expected[1]. Thus, we proceed by calculating the band structure of Au/GaAs(111) with SOC. As shown in Fig. 4a, an SOC gap is opened at the $\Gamma$ point; the orbital components for band I, II and III remain the same as that without SOC (Fig. 3b). Because of the large SOC in Au and GaAs, the size of SOC gap is very significant of $\sim 73$ meV (Fig. 4b), which facilitates possibly the room-temperature measurement. In addition, the inversion symmetry is broken in Au/GaAs(111), so that the band degeneracy is lifted by Rashba splitting. Neglecting the Rashba effect, the SOC gap and bands can be well reproduced by including an on-site SOC term in our minimal-basis tight-binding band V, as shown in Fig. 1g. A further

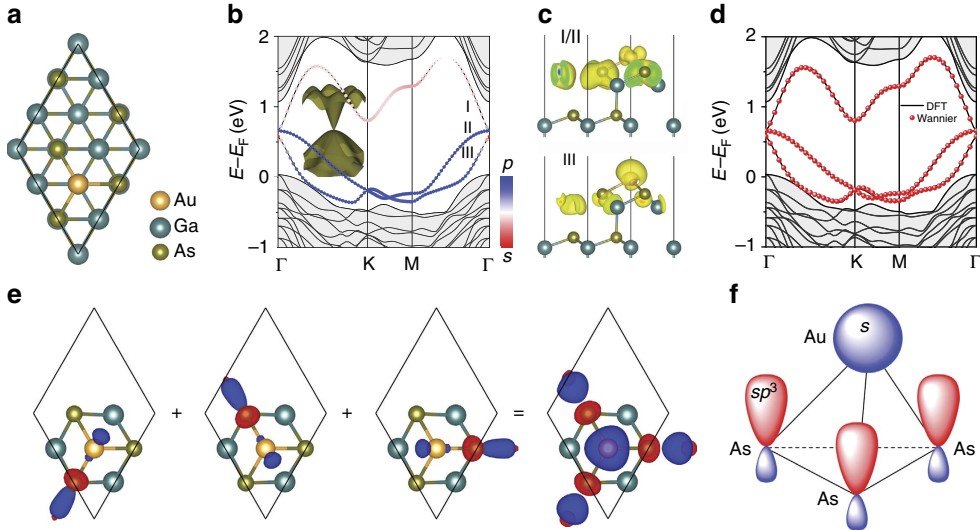

**Figure 3 | Band and orbital analysis for Au/GaAs(111) without SOC.** (**a**) Top view of $\sqrt{3}\times\sqrt{3}R30°$ superlattice structure for Au grown on As-terminated GaAs(111) surface. (**b**) Band structure of Au/GaAs(111) superlattice without SOC. The inset is 3D plotting of I, II and III bands around $\Gamma$ point. The red and blue colours indicate the component of $s$ and $p$ orbitals, respectively. (**c**) Charge density distribution of I, II and III bands at $\Gamma$ point, showing the surface character. (**d**) Comparison between density functional theory (DFT) bands (solid lines) and MLWFs fitted bands (red dots). (**e**) Top view of three MLWFs fitted from the DFT bands and the overall orbital shape by adding them together. Red and blue colours denote positive and negative value, respectively. (**f**) Schematic view of the un-hybridized orbitals. One $s$ orbital for Au and three $sp^3$ orbitals for three As atoms, forming a tetrahedron structure. For clarity, only Au and the top two (one) bilayer GaAs atoms are plotted in **c**,**e**.

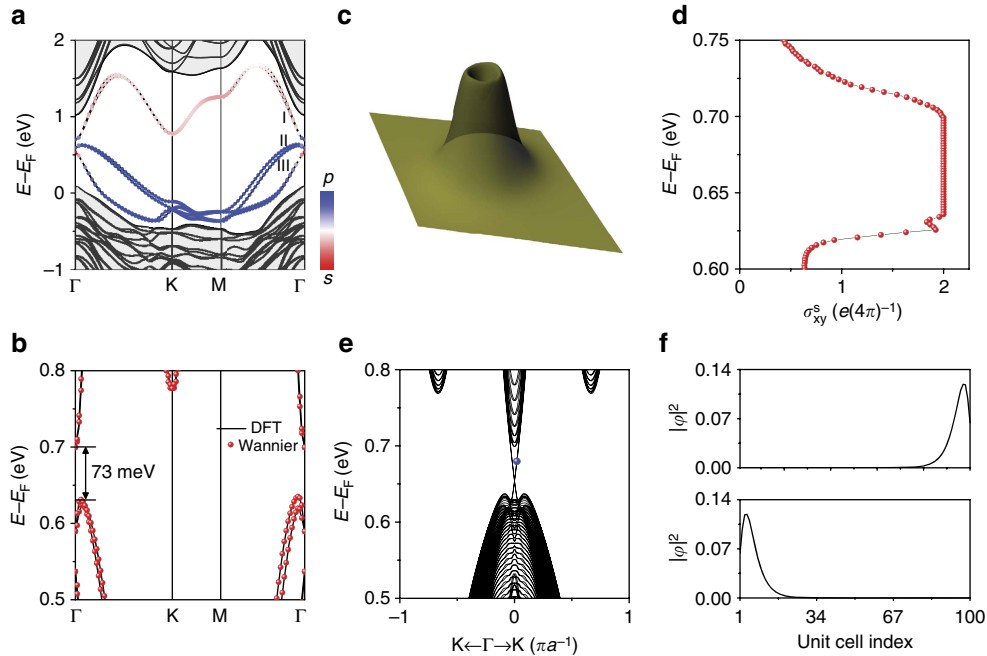

**Figure 4 | Non-trivial topological phase for Au/GaAs(111) with SOC.** (**a**) Band structure of Au/GaAs(111) superlattice with SOC. The red and blue colours indicate the component of $s$ and $p$ orbitals, respectively. (**b**) Zoom-in comparison between density functional theory (DFT) bands (solid lines) and MLWFs fitted bands (red dots) around the SOC bandgap. (**c**) Spin Berry curvature around the $\Gamma$ point by setting the Fermi level within the energy window of SOC gap. (**d**) Spin Hall conductance as a function of Fermi level, showing quantized value within the energy window of SOC gap. (**e**) 1D ribbon band structure, showing gapless Dirac edge states within the energy window of SOC gap. (**f**) Real space distribution of the Dirac edge states at the energy marked by a blue dot in **e**. The two degenerate edge states are localized at opposite (left and right) edge of the ribbon.

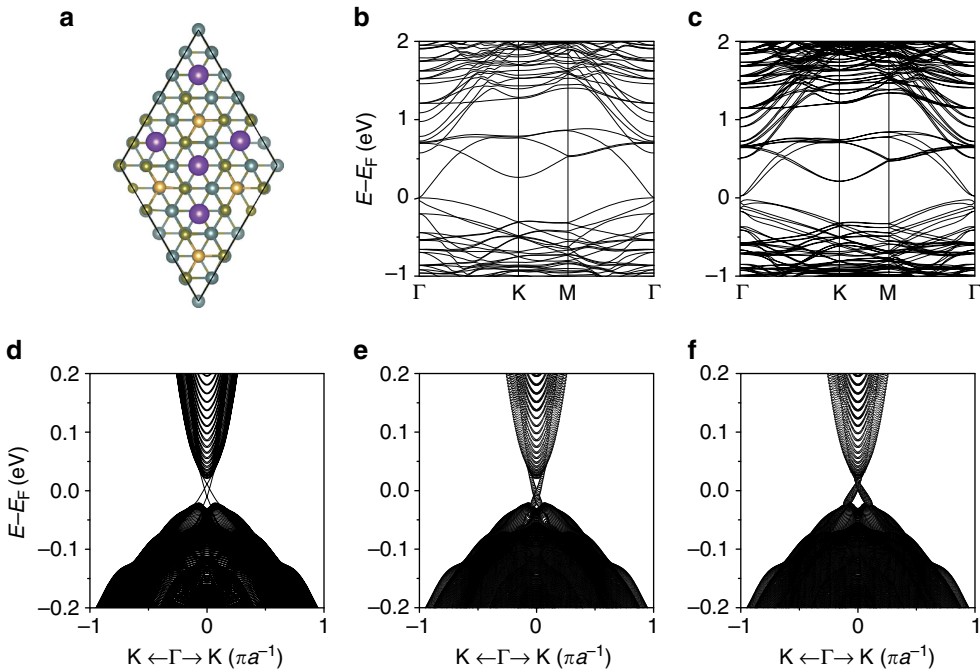

**Figure 5 | Intrinsic 2D QSH for Au/GaAs(111) with surface $n$-doping.** (**a**) Atomic structure of Au/GaAs(111) at 5/12 coverage of K atoms. (**b**,**c**) Band structures without and with SOC. (**d**) 1D ribbon band structure, showing gapless Dirac edge states within the energy window of SOC gap. (**e**,**f**) 1D ribbon band structure projected onto edge unit cell, showing opposite (left and right) edge state, respectively. In the projected bands, circle size denotes the weighting factor of the corresponding states.

comparison between the fitted Wannier band and first-principles band is shown in Fig. 4b, showing excellent agreement.

To clearly identify the non-trivial topology in Au/GaAs(111), we have calculated its spin Berry curvature, spin Hall

conductance and edge states by using the fitted Wannier Hamiltonian. The spin Hall conductance ($\sigma_{xy}^{s} = 2C_{s} \cdot \frac{e}{4\pi}$) is defined based on the spin Chern number ($C_{s}$), which can be calculated from the spin Berry curvature by using standard Kubo

formula[10]. Here $e$ is the elementary charge and $\pi$ is the mathematical constant. Figure 4c shows the spin Berry curvature around the $\Gamma$ point, which is comparable to that shown in Fig. 2f. Figure 4d shows the calculated spin Hall conductance as a function of Fermi level, which has a quantized value within the energy window of SOC gap, demonstrating the critical feature of QSH phase. The non-zero spin Chern number can also be manifested by the presence of gapless edge states within the SOC gap. One-dimensional (1D) ribbon band structure of Au/GaAs(111) with a width of 100 unit cells are calculated[33], as shown in Fig. 4e. A pair of gapless edge states with a Dirac cone at the $\Gamma$ point, which are energy-degenerate for opposite (left and right) edges, are seen within the SOC gap. The real space distribution of the edge states, at the energy marked by blue dot in Fig. 4e, is shown in Fig. 4f. Clearly, the degenerated edge states are spatially localized at the opposite edges (left and right) of the ribbon. The number of the edge states indicates the absolute value of the spin Chern number, which is consistent with the calculated $C_s = 1$ in Fig. 4d.

**Surface $n$-doping to tune Fermi level.** We note that the Fermi level of Au/GaAs(111) is not in the non-trivial SOC gap, so $n$-doping is needed. Generally, $n$-doping can be more easily realized than $p$-doping in surface systems. For example, alkali-metal atoms have been widely used for surface $n$-doping in the experiments, including graphene[34], superconducting FeSe (ref. 35) and metal film[30]. Thus, we have adopted the same strategy using K to $n$-dope the Au/GaAs(111) surface. Our calculations show that K prefers to adsorb at the hexagonal close-packed hollow site and form a uniform distribution due to the Coulomb repulsion between charged K atoms[15]. At 1/3 and 2/3 monolayer (ML) of K coverage, the Fermi level of Au/GaAs(111) can be tuned close to and below and above the SOC gap, respectively, and 1D ribbon calculations further demonstrate their non-trivial topological edge states (Supplementary Fig. 1). The Fermi level can be continuously moved upward, so at 5/12 ML (see atomic structure in Fig. 5a modelled by a larger $2 \times 2$ supercell), the Fermi level is moved into the Dirac band without SOC (Fig. 5b) or inside the gap with SOC (Fig. 5c). It is important to notice that K surface adsorption provides an ideal $n$-doping by moving the Fermi level without affecting the bands of host system in all the cases studied. Furthermore, 1D ribbon calculations demonstrate two non-degenerated gapless Dirac edge states within SOC gap, as shown in Fig. 5d. From the edge-projected bands, the left (Fig. 5e) and right (Fig. 5f) Dirac edge states can be distinguished. Thus, Au/GaAs(111) can be tuned into an intrinsic 2D TI with surface alkali-metal doping. Moreover, we have done molecular dynamics simulations in a canonical assemble at room temperature to further demonstrate the stability of K atoms on GaAs surface (Supplementary Fig. 2).

## Discussion

Currently, there are two experimental methods that can be used to detect topological edge states in 2D TIs. One is the transport measurement to measure the quantized conductance induced by topological edge state, and the other is scanning tunnelling spectroscopy measurement to image the real-space topological edge state within the energy window of SOC gap. For the second method, even if the Fermi level is below the non-trivial SOC gap (a non-intrinsic TI), its topological edge state can still be observed experimentally[36–38], because the empty state of TI far away from the Fermi level can be easily accessed by scanning tunnelling spectroscopy[39]. Thus, we believe our work may stimulate new experimental studies of Au/GaAs(111) beyond previous efforts, especially for K-doped systems. Without our theoretical

prediction, however, this system could be easily overlooked because it has a trigonal (Au) lattice, which is not known to support topological state before.

In summary, we introduce a generic minimal-basis three-orbital QSH model in trigonal lattice, which can be reduced to an effective two-band model at the continuum limit. Furthermore, we suggest possible experimental realization of our model in the Au/GaAs(111) system that has already been experimentally grown previously. Our findings not only enrich the basic knowledge of QSH phase but also extend the search for QSH materials to new lattice and orbital types. We believe our proposed new classes of QSH materials can be generalized to other metal elements and semiconductor surfaces for experimental exploration.

## Methods

**The first-principles calculations.** The first-principles calculations are carried out in the framework of generalized gradient approximation with Perdew–Burke–Ernzerhof functionals using the Vienna Ab initio simulation package[40]. The $\sqrt{3} \times \sqrt{3}R30°$ GaAs(111) surface is simulated by six-bilayer slabs with lattice constant $a = 7.06$ Å obtained from the optimized bulk GaAs. The bottom surface is saturated by pseudo-hydrogen atoms fractionally charged with $1.25e$. All the calculations are performed with a plane-wave cutoff of 400 eV on the $9 \times 9 \times 1$ Monkhorst–Pack $k$-point mesh. The vacuum layer of 15 Å thick is used to ensure decoupling between neighbouring slabs. During structural relaxation, bottom two-bilayer GaAs and hydrogen atoms are fixed. The Au and top four-bilayer GaAs atoms are relaxed until the forces are smaller than 0.01 eV Å$^{-1}$.

**Data availability.** The data that support the findings of this study are available from the corresponding author on request.

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

## Acknowledgements

Z.F.W. acknowledges financial support from Chinese Youth 1000 Talents Program and Fundamental Research Funds for the Central Universities. K.-H.J. and F.L. acknowledge financial support from DOE-BES (No. DE-FG02-04ER46148). We also thank Supercomputing Center at USTC, NERSC and CHPC at University of Utah for providing the computing resources.

## Author contributions

F.L. and Z.F.W. conceived the project; K.-H.J. and Z.F.W. performed tight-binding model calculations; Z.F.W. performed first-principles calculations; Z.F.W., K.-H.J. and F.L. prepared the manuscript.

## Additional information

**Competing financial interests:** The authors declare no competing financial interests.

**How to cite this article**: Wang, Z. F. *et al.* Quantum spin Hall phase in 2D trigonal lattice. *Nat. Commun.* 7:12746 doi: 10.1038/ncomms12746 (2016).

