## [Peer review file · Nature Communications]

Reviewers' comments:

Reviewer #1 (Remarks to the Author):

Here, the authors presented a theoretical study of quantum spin Hall phase in two-dimensional trigonal lattice. Their TB model analyses show there is a SOC-induced band inversion at the G point. And this exact model of QSH phase was shown to be realizable in an experimental system of Au/GaAs(111) surface by using first-principles calculations. They claimed that the QSH phase in this system can be readily confirmed.

For the Au/GaAs(111) surface, the authors mentioned that this system is easier to facilitate experimental measurement. I can not buy this argument. The argument would be truly exciting if the Fermi level of the proposed structure lies in its band gap. Unfortunately, we can see the Fermi level lies far away (0.6 eV) from the nontrivial band gap. To be a truly 2D TI, we have to shift the Fermi level up by electron-doping or something else. But I do not think it is so easy to do that. That is also why Au/GaAs(111) surface has been realized in experiment, but its topological properties has not been observed yet.

In fact, such systems have already been extensively studied in many previous works. For example, Nano Lett., 15 (10), 6568, 2015; PRB 92, 081112(R) (2015). Compared with the present work, I believe the QSH effect is easier to be observed in these previous systems as the Fermi level of these systems lies exactly in their band gap. That is very importantly. Besides, these previous related works are not well reflected in the present paper.

Without turning on SOC, there is a Dirac point above the Fermi level in Au/GaAs(111) surface. To demonstrate that it is truly Dirac point, it is better to show its 3D plot.

Considering these facts, I do not think that it meets standards of NC and thus I can not recommend the publication of the present paper.

Reviewer #2 (Remarks to the Author):

In the paper "Quantum Spin Hall Phase in 2D Trigonal Lattice: Minimal Basis Model and Material Prediction", Wang et al. proposed a theoretical model to use the trigonal lattice for the realization of quantum spin Hall (QSH) effect, by the coupling of three bands containing s, p_x and p_y orbitals. From their DFT calculations and tight-binding model analyses, they found that the QSH phase is induced by the spin-orbit coupling from the p orbitals. Moreover, the authors found a real system, i.e., Au/GaAs(111) surface with a large energy gap, for the experimental verification. The work is a meaningful, though is not totally new in this context. Since the realization of the QSH effect as predicted in the Kane-Mele model is still a challenge, particularly at room temperature, this paper merits publication in Nat. Comm. Overall, this paper was well written. Some points and descriptions can be further improved. For example, in Fig. 4(f), authors plot two edge states, but in Fig. 4(e), the authors just describe one state, the authors should add another one.

Reviewer #3 (Remarks to the Author):

This manuscript reported by Wang et al. theoretically studied the quantum spin Hall (QSH) phase in a two-dimensional trigonal lattice, and found a minimal basis of three orbitals to produce a QSH phase. Based on first-principles calculations, they also predicted Au/GaAs(111) system could realize this QSH phase with the exact minimal model.

From the first theoretical proposal of QSH, only two materials systems (HgTe/CdTe quantum wells and InAs/GaSb quantum wells) with a tiny band gap have been experimentally confirmed under

extremely low temperature (mK), which severely restricts potential applications in spintronics of the QSH effect. Therefore it is desirable to find new realizable QSH systems with excellent performance. The authors made great effort in this direction. However, I have comments on both the effective models and the Au/GaAs(111) candidate, seen in the following.

For the effective models:

In my opinion, the BHZ model is quite general to describe the QSH, and it is not just limited to be used in square lattices. Therefore, it might not be appropriate that the authors call BHZ model as just quantum well models in the first paragraph of page 2.

The first-type and second-type QSH phases of this three-band model seem to have the same essence of the s-p-type band inversion, which could be described by the BHZ model. For the second-type QSH phase, the authors should not throw the s orbital to obtain the effective two-band model.

Especially, for the second type QSH phase, the effective two-band model from the three-band model might be topologically trivial, by considering the minus parties at all TRIMs. Is it right? I suggest the authors to check its topological property of this two-band model. Can the authors calculate the surface states based on the two-band model?

For the candidate of Au/GaAs(111) system:

I notice that the chemical potential in Au/GaAs(111) system is far from the band gap around the Gamma point, so this phase is essentially a metal phase, not an insulator phase. In this sense, Au/GaAs(111) surface is not a real QSH phase.

In summary, I could not recommend publishing this manuscript with the current version.

Response to Reviewers' comments:

We are very thankful to all three reviewers whose comments are very constructive, insightful and helped us to further improve the quality of our work and clarify some technical points in the MS. We believe we have adequately addressed all the questions raised by the reviewers, as detailed in our point-to-point reply below. We sincerely hope that all three reviewers will now be convinced to recommend our work for publication in Nature Communications.

Summary of changes:

All the revised and newly added texts are written in “BLUE” color to indicate changes. In the main MS, we have added a new Fig. 5 for alkali-metal n-type doping and related discussion. In the supplementary information, we have added a new Fig. S1.

Response to Reviewer #1

Question 1: Here, the authors presented a theoretical study of quantum spin Hall phase in two-dimensional trigonal lattice. Their TB model analyses show there is a SOC-induced band inversion at the G point. And this exact model of QSH phase was shown to be realizable in an experimental system of Au/GaAs(111) surface by using first-principles calculations. They claimed that the QSH phase in this system can be readily confirmed. For the Au/GaAs(111) surface, the authors mentioned that this system is easier to facilitate experimental measurement. I cannot buy this argument. The argument would be truly exciting if the Fermi level of the proposed structure lies in its band gap. Unfortunately, we can see the Fermi level lies far away (0.6 eV) from the nontrivial band gap. To be a truly 2D TI, we have to shift the Fermi level up by electron-doping or something else. But I do not think it is so easy to do that. That is also why Au/GaAs(111) surface has been realized in experiment, but its topological properties has not been observed yet.

Reply: We thank the reviewer for bringing up this important point. Yes, we agree that without showing an intrinsic TI system with the Fermi level lying inside the nontrivial SOC gap, our work may fall short for publication in Nature Communications. So, we have done some further literature search and thinking, and fortunately we found that n-type doping of 2D system on a substrate as we propose here is very feasible (much more so than p-type doping) and have already been experimentally demonstrated in several different systems. Specifically, n-doping of a surface layer has been achieved by growth of alkali metals, such as K, in experiments, including graphene [Nature Commun. 5, 3257 (2014)], superconducting FeSe [Nature Mater. 14, 775 (2015)] and metal film [Sci. Rep. 4, 4742 (2014)]. The most important aspect is that K deposition provides an ideal doping scenario, only shifting the Fermi level without affecting the electronic bands of host system. Therefore, we have adopted and tested this strategy for our case, with all the new calculation results added. Specifically, for Au/GaAs(111) surface, our calculations show that K prefers to adsorb at the hcp hollow site and form a uniform distribution due to the Coulomb repulsion between charged K atoms. At 1/3 ML [Fig. R1(a)] and 2/3 ML [Fig. R1(d)] coverage of K, the Fermi-level of Au/GaAs(111) can be tuned close to and below [Fig. R1(b)] and above [Fig. R1(e)] the SOC gap, respectively. The 1D ribbon calculations further demonstrate their nontrivial topological edge states, as

shown in Fig. R1(c) and Fig. R1(f), respectively. The Fermi level can be tuned continuously by increasing the K coverage, so at 5/12 ML, as obtained with a calculation using a larger 2x2 supercell of Au/GaAs(111) [Fig. R2(a)], the Fermi-level is moved into the SOC gap, as shown in Fig. R2(b) and R2(c). Furthermore, 1D ribbon calculations demonstrate two non-degenerated nontrivial Dirac edge states [Fig. R2(d)] within the SOC gap. From the edge projected bands, we can clearly distinguish these two Dirac edge states for left and right edge [Fig. R2(e) and R2(f)], respectively. Therefore, Au/GaAs(111) can be tuned into an intrinsic 2D TI with surface alkali-metal doping.

Figure R1. Band structure and topological edge state for Au/GaAs(111) with surface K doping. a,d, Atomic structure of Au/GaAs(111) with 1/3 and 2/3 coverage of K atoms. b,e, Band structure of a and d with SOC. c,f, 1D ribbon band structure of a and d, showing the gapless Dirac edge states within the energy window of SOC gap..

Figure R2. Intrinsic 2D QSH for Au/GaAs(111) with surface K doping. a, Atomic structure of Au/GaAs(111) with 5/12 coverage of K atoms. b,c, Band structure without and with SOC. d, 1D ribbon

band structure, showing gapless Dirac edge states within the energy window of SOC gap. **e,f**, 1D ribbon band structure projected onto edge unit cell, showing left and right edge state, respectively. In the projected bands, circle size denotes the weighting factor of the corresponding states

We have added the above new n-doping results as Fig. 5 with the following discussions in the revised MS:

“We note that the Fermi-level of Au/GaAs(111) is not in the non-trivial SOC gap, so n-doping is needed. Generally, n-doping can be more easily realized than p-doping in surface systems. For example, alkali-metal atoms have been widely used for surface n-doping in the experiments, including graphene³⁴, superconducting FeSe³⁵ and metal film³⁰. Thus, we have adopted the same strategy using K to n-dope the Au/GaAs(111) surface. Our calculations show that K prefers to adsorb at the hcp hollow site and form a uniform distribution due to the Coulomb repulsion between charged K atoms¹⁵. At 1/3 and 2/3 ML of K coverage, the Fermi-level of Au/GaAs(111) can be tuned close to and below and above the SOC gap, respectively, and 1D ribbon calculations further demonstrate their non-trivial topological edge states (see Fig. S1 in Supplementary Information). The Fermi-level can be continuously moved upward with increasing K coverage, so at 5/12 ML (see the corresponding atomic structure in Fig. 5a modeled by a 2×2 supercell), the Fermi-level is moved into the Dirac band without SOC (Fig. 5b) or inside the gap with SOC (Fig. 5c). It is important to notice that K surface adsorption provides an ideal n-doping by moving the Fermi level without affecting the bands of host system in all the cases studied. Furthermore, 1D ribbon calculations demonstrate two non-degenerated gapless Dirac edge states within SOC gap, as shown in Fig. 5d. From the edge projected bands, the left (Fig. 5e) and right (Fig. 5f) Dirac edge states can be distinguished. Thus, Au/GaAs(111) can be tuned into an intrinsic 2D TI with surface alkali-metal doping.”

On the other hand, we would like to clarify one more point here. Currently, there are two experimental methods which can be used to detect the topological edge states in 2D TI materials. One is the direct transport measurement to detect the quantized conductance induced by topological edge states, and the other is scanning tunneling spectroscopy (STS) measurement to image real-space topological edge state within the energy window of SOC gap. For the second method, even if the Fermi-level is below the nontrivial SOC gap (a non-intrinsic TI), its topological edge state can still be observed experimentally [Nature Phys. 10, 664 (2014)], because the empty state of TI far away from the Fermi-level can be easily accessed by STS measurement [Phys. Rev. B 91, 161306(R) (2015)]. Therefore, we think that the reviewer’s statement *“For the Au/GaAs(111) surface, the authors mentioned that this system is easier to facilitate experimental measurement. I cannot buy this argument. ... This is also why Au/GaAs(111) surface has been realized in experiment, but its topological properties has not been observed yet”* is not fully warranted, because even without doping the topological nature we predicted in Au/GaAs(111) could still be observed by STS. Of course, such experiment has not been and might never be performed without knowing the system is topological non-trivial as our theory predicted here, especially people may easily overlook a system with trigonal lattice that was not known before to support topological state. Thus, we believe our work will likely stimulate new experimental studies of Au/GaAs(111) beyond previous efforts,

especially for K-doped systems. For the above reasons, we originally focused on developing a generic model for QSH phase in trigonal lattice (NEW) and identify a material system to realize it (experimentally realized already as the reviewer pointed out), but overlooked the problem of Fermi level position). In this regard, we greatly appreciate the reviewer's comment which motivated us to perform the additional calculations of K-doping that may also facilitate direct quantum transport measurement on our proposed system. We have added the following statements to clarify this point:

“On the other hand, there are two experimental methods which can be used to detect topological edge states in 2D TIs. One is transport measurement to measure the quantized conductance induced by topological edge state, and the other is scanning tunneling spectroscopy (STS) measurement to image the real-space topological edge state within the energy window of SOC gap. For the second method, even if the Fermi-level is below the non-trivial SOC gap (a non-intrinsic TI), its topological edge state can still be observed experimentally³⁶, because the empty state of TI far away from the Fermi-level can be easily accessed by STS³⁷. Thus, we believe our work may stimulate new experimental studies of Au/GaAs(111) beyond previous efforts, especially for K-doped systems to enable quantum transport measurement. Without our theoretical prediction, however, this system would be easily overlooked because it has a trigonal (Au) lattice which is not known to support topological state before”

Question 2: In fact, such systems have already been extensively studied in many previous works. For example, Nano Lett., 15 (10), 6568, 2015; PRB 92, 081112(R) (2015). Compared with the present work, I believe the QSH effect is easier to be observed in these previous systems as the Fermi level of these systems lies exactly in their band gap. That is very importantly. Besides, these previous related works are not well reflected in the present paper.

Reply: We would like to bring to reviewer's attention a recent invited review article (arXiv:1509.09016v1) by Yafei Ren, Zhenhua Qiao and Qian Niu, which will appear soon in Reports on Progress in Physics. They provided a comprehensive (likely the most complete) list of 2D TI systems that have been studied so far, and one will find that among all the real material systems they listed, most have the hexagonal lattice (including the two papers mentioned above by the reviewer which we have added citation to them) and a few have square lattice. This clearly indicates the “trigonal lattice” we propose here is NEW (different from previous lattice type) and significant, because it will greatly broaden the scope of 2D TI systems especially considering the fact many metal systems prefer the closed-packing trigonal structure as pointed out in our MS. On the other hand, we agree it is critical to predict an intrinsic TI (like the ones mentioned by the reviewer), which we now addressed by additional doping calculations of our system discussed above.

Question 3: Without turning on SOC, there is a Dirac point above the Fermi level in Au/GaAs(111) surface. To demonstrate that it is truly Dirac point, it is better to show its 3D plot.

Reply: We thank the reviewer for this nice suggestion. We have now added a 3D plot to show a truly Dirac point in revised Fig. 3, as duplicated below in Fig. R3.

Figure R3. 3D band plotting for the three bands (I, II and III) in Fig. 3, showing the truly Dirac point.

Question 4: Considering these facts, I do not think that it meets standards of NC and thus I cannot recommend the publication of the present paper.

Reply: Once again, we thank the reviewer for his critical, constructive and helpful comments. We believe we have addressed all the questions raised by the reviewer and trust that he will now recommended our MS for publication in Nature Communications.

Response to Reviewer #2

Question 1: In the paper "Quantum Spin Hall Phase in 2D Trigonal Lattice: Minimal Basis Model and Material Prediction", Wang et al. proposed a theoretical model to use the trigonal lattice for the realization of quantum spin Hall (QSH) effect, by the coupling of three bands containing s, p_x and p_y orbitals. From their DFT calculations and tight-binding model analyses, they found that the QSH phase is induced by the spin-orbit coupling from the p orbitals. Moreover, the authors found a real system, i.e., Au/GaAs(111) surface with a large energy gap, for the experimental verification. The work is a meaningful, though is not totally new in this context. Since the realization of the QSH effect as predicted in the Kane-Mele model is still a challenge, particularly at room temperature, this paper merits publication in Nat. Comm.

Reply: We are very pleased that the reviewer recommended our MS for publication in Nature Communications. To further stress the novelty of our work, we refer the reviewer to our response to reviewer 1's Question#2, if necessary.

Question 2: Overall, this paper was well written. Some points and descriptions can be further improved. For example, in Fig. 4(f), authors plot two edge states, but in Fig. 4(e), the authors just describe one state, the authors should add another one.

Reply: Thanks for the suggestion. Since the two edge states are energy-degenerate for the 1D ribbon band structure in Fig. 4(e), we didn't distinguish them in our previous MS. To clarify this point, we have made the following revision.

"A pair of gapless edge states with a Dirac cone at the Γ point, which are energy-degenerate for opposite (left and right) edges, are seen within the SOC gap. The real space distribution of the edge states, at the

energy marked by blue dot in Fig. 4e, is shown in Fig. 4f. Clearly, the degenerated edge states are spatially localized at the opposite edges (left and right) of the ribbon.”

Response to Reviewer #3

Question 1: This manuscript reported by Wang et al. theoretically studied the quantum spin Hall (QSH) phase in a two-dimensional trigonal lattice, and found a minimal basis of three orbitals to produce a QSH phase. Based on first-principles calculations, they also predicted Au/GaAs(111) system could realize this QSH phase with the exact minimal model. From the first theoretical proposal of QSH, only two materials systems (HgTe/CdTe quantum wells and InAs/GaSb quantum wells) with a tiny band gap have been experimentally confirmed under extremely low temperature (mK), which severely restricts potential applications in spintronics of the QSH effect. Therefore it is desirable to find new realizable QSH systems with excellent performance. The authors made great effort in this direction. However, I have comments on both the effective models and the Au/GaAs(111) candidate, seen in the following.

Reply: We thank the reviewer for considering our work to be a great effort in search of realistic QSH material.

Question 2: For the effective models: In my opinion, the BHZ model is quite general to describe the QSH, and it is not just limited to be used in square lattices. Therefore, it might not be appropriate that the authors call BHZ model as just quantum well models in the first paragraph of page 2.

Reply: We agree with the reviewer that BHZ model is general in describing different QSH systems if one considers only in terms of band inversion mechanism. However, BHZ model is an effective model that is only valid for describing the band around the band inversion k-point and it was originally derived from a square lattice for HgTe quantum well system [Science 314, 1757 (2006)]. To clarify this point, we made the following revised statements in our MS.

“Currently, there are two prevailing theoretical models for QSH phase in a 2D system: the Kane-Mele (KM) model¹ and the Bernevig-Hughes-Zhang (BHZ) model^{2,3}. In KM model, the QSH phase is realized by any finite SOC induced band gap opening at Dirac point, as a generalization of Haldane’s model⁴ to spinful system with time reversal symmetry in a hexagonal lattice. In BHZ model, the QSH phase is realized by an SOC induced band inversion at time reversal invariant momenta (TRIM) between two bands of different parities, originally derived from a square lattice of HgTe quantum well system.”

Question 3: The first-type and second-type QSH phases of this three-band model seem to have the same essence of the s-p-type band inversion, which could be described by the BHZ model. For the second-type QSH phase, the authors should not throw the s orbital to obtain the effective two-band model. Especially, for the second type QSH phase, the effective two-band model from the three-band model might be topologically trivial, by considering the minus parties at all TRIMs. Is it right? I suggest the authors to check its topological property of this two-band model. Can the authors calculate the surface states based on the two-band model?

Reply: We apologize for the confusion about our continuum model for the second-type QSH phase. First, we point out that the parity at the TRIMs (Fig. 1f and 1g) cannot be used to distinguish the topological phase in our two-band continuum model. As shown by our derivation in supplementary information, in reducing the three-band to two-band model, the s orbital is not simply removed, so the remaining two orbitals are not pure p components but the mixed states of both s and p orbitals at the two-band continuum limit. Consequently, we cannot write out the mathematical form of orbital basis at the continuum limit. This is distinctively different from the original three-band model, in which s and p orbitals are decoupled at the TRIMs. Therefore, it is inappropriate to define the band parity around Γ point in our two-band continuum model ($s+p$, neither odd nor even parity). Nevertheless, the most important feature in the two-band continuum model is the SOC gap opened at the Dirac point, indicating a non-trivial topology. Furthermore, the non-trivial topology can be characterized from the two-band continuum model by plotting the non-zero Berry curvature around Γ point, as shown in Fig. 2f, which can be directly compared with our DFT berry curvature shown in Fig. 4c. The good agreement between the two confirms qualitatively the validity of the two-band model. However, it is hard (if not impossible) to plot a ribbon band structure using the two-band continuum model, because it is unclear how to map the model back onto a chosen lattice; or in other words, as a continuum model, it does not belong to a defined lattice type. All the lattice models (such as BHZ model starting from a square lattice) can in principle be adiabatically reduced to a continuum model, but the reverse is not true. This is also the reason why people do only Berry curvature calculations for such continuum model in previous theoretical works. Finally, to address the reviewer's question in a broad perspective, we emphasize that the importance of developing new type of lattice models, as we have done here, is not only to enrich the physics of topological theory (for example, according to an expert of field theory whom we have discussed with, there remains a fundamental question to be answered: "is there a unified theory to adiabatically reduce all the lattice-type models to the common continuum limit?"), but also to stimulate search of topological states in new classes of materials (such as 2D materials with trigonal lattice we show here).

Question 4: For the candidate of Au/GaAs(111) system: I notice that the chemical potential in Au/GaAs(111) system is far from the band gap around the Gamma point, so this phase is essentially a metal phase, not an insulator phase. In this sense, Au/GaAs(111) surface is not a real QSH phase. In summary, I could not recommend publishing this manuscript with the current version.

Reply: We thank the reviewer for pointing out this important point, please see our reply to Question #1 by reviewer 1, along with our newly added doping calculations. In summary, we believe that we have addressed all the questions raised by the reviewer, and trust that he/she is now convinced to recommend our manuscript for publication in Nature Communications.

REVIEWERS' COMMENTS:

Reviewer #1 (Remarks to the Author):

Thanks for the authors' prompt reply and additional calculations. Although the results are novel and potentially exciting, I am still not convinced that this paper meets standards of Nature Communications. I think this paper fits better a more specialized journal like PRB or NJP. Besides, the authors should consider the following issues:

(1) Yes, the n-doping of a surface layer can be realized in experiments and there are many avenues to get it. But that does not mean that this can be "easily" realized in experiments. The results would be truly exciting if the proposed structures were easy to make experimentally. For practical application, we must consider the "final cost". There are many many promising 2D TIs that have been proposed. And obviously, compared with those works, this system studied here is not so interesting as claimed by the authors. For this system, first we need to grow this theoretically designed system, I think the well-controlled techniques are necessary. After that, we even need to employ some more complex techniques to tune the Fermi level. So this system is not "promising" at all.

(2) For simulating the growth of alkali metals, did the authors consider the vdw interactions? This is very important, but I did not see any statement on this point. For the corresponding structures, the details should be listed in the supporting information. And for K atoms on the surface, are they stable? I mean at room temperature. The authors should demonstrate it.

(3) In the revised version, the authors still did not include a discussion on the difference between their work and the previous similar works [Nano Lett., 15 (10), 6568, 2015; PRB 92, 081112(R) (2015)]. They claimed that compared with those works, their system is new because of "trigonal lattice". It makes no sense. For those previously proposed structures, QSH effect is easier to be observed as the Fermi level lying in the band gap.

Reviewer #3 (Remarks to the Author):

The authors well addressed my questions and correspondingly modified their manuscript. I am satisfied with their response. Now I recommend publishing it in Nature Communications.

Response to Reviewers' comments:

We thank all three reviewers for reviewing our MS. We are very pleased that two out of three reviewers have recommended publication of our MS in Nature Communications. However, we respectively disagree with the reviewer #1's overall evaluation of our work as explained below.

Response to Reviewer #1

Question 1: Thanks for the authors' prompt reply and additional calculations. Although the results are novel and potentially exciting, I am still not convinced that this paper meets standards of Nature Communications. I think this paper fits better a more specialized journal like PRB or NJP. Besides, the authors should consider the following issues.

Yes, the n-doping of a surface layer can be realized in experiments and there are many avenues to get it. But that does not mean that this can be "easily" realized in experiments. The results would be truly exciting if the proposed structures were easy to make experimentally. For practical application, we must consider the "final cost". There are many many promising 2D TIs that have been proposed. And obviously, compared with those works, this system studied here is not so interesting as claimed by the authors. For this system, first we need to grow this theoretically designed system, I think the well-controlled techniques are necessary. After that, we even need to employ some more complex techniques to tune the Fermi level. So this system is not "promising" at all.

Reply: We found that some of the reviewer's comments are self-contradicting. For example, he first stated "the results are novel and potentially exciting", but then said "this system is not 'promising' at all", which is confusing.

We would like to reiterate again that the main points of our paper is to demonstrate a theoretical model of topological phase with minimum basis in a "trigonal" lattice as well as a real material system for its realization. Our work not only presents "new physics", but also offers a "promising" route towards experimental discovery of new 2D topological insulators, because it is well known that metal elements prefer a close-packing "trigonal" 2D lattice over an open lattice like hexagonal lattice, which is employed in most if not all previous theoretical studies. There are already ample experimental cases of growing the close-packing trigonal structure on a substrate through epitaxial growth process, including the exact system of Au grown on GaAs (111) substrate we showed in the MS. Furthermore, deposition of alkali atoms for doping the overlayer on a substrate has also been demonstrated by several experiments as we referenced in the paper (Refs. [30,34,35]). In contrast, despite many 2D hexagonal lattice structures have been proposed theoretically, either freestanding or on a substrate, none of them except Bi(111) with the desired topological properties has been shown in existing experimental literature. These evidence supports that our proposed "trigonal" system is promising, or at least promising as an alternative to the previous hexagonal

systems.

Question 2: For simulating the growth of alkali metals, did the authors consider the vdw interactions? This is very important, but I did not see any statement on this point. For the corresponding structures, the details should be listed in the supporting information. And for K atoms on the surface, are they stable? I mean at room temperature. The authors should demonstrate it.

Reply: The physical origin of vdw interaction is dispersive force arising from polarizability. Therefore, vdw interactions are generally not important for alkali atoms, especially in ionized form, because they are not polarizable. In comparison, vdw interactions are important for rare gas atoms, negative ions, and molecules. Anyway, only for this reviewer's information, we have done calculations of band structures with vdw functional (see Fig. R1 below), which are same with those with standard DFT (Fig. 4 in the paper). This is expected; otherwise, one would question the validity of the vdw functional used.

We have done MD simulations at RT to show the stability of the proposed surface structures (see Fig. R2 below). On the other hand, so far the only two 2D TI (QSH) systems confirmed by experiments are done at temperatures of milli Kelvin. So a room temperature TI would be great, but presently even an experiment at a temperature of a few tens of Kelvin will be a significant advancement.

Figure R1. Band structures with Van der Waals (vdw) interaction. a,b Band structures of Au/GaAs(111) at 5/12 coverage of K atoms without and with SOC. The band features are the same as those shown in Fig. 4 without vdw interaction.

Figure R2. Stability of K atoms on Au/GaAs(111) surface. a, Temperature variation with time, and the open blue dots denote the times of snap shots taken in b-g. The first-principles Molecular dynamics (MD) simulations for Au/GaAs(111) at 5/12 coverage of K atoms are done using NVT ensemble at 300 K. b-g, Snap shots of the structures at different times, demonstrating the stability of surface K atoms at room temperature.

Question 3: In the revised version, the authors still did not include a discussion on the difference between their work and the previous similar works [Nano Lett., 15 (10), 6568, 2015; PRB 92, 081112(R) (2015)]. They claimed that compared with those works, their system is new because of "trigonal lattice". It makes no sense. For those previously proposed structures, QSH effect is easier to be observed as the Fermi level lying in the band gap.

Reply: We did already make a comparison and explained the advantage of our proposed 'trigonal' lattice over the 'hexagonal' lattice, as reiterated in our response to reviewer's first comment. We believe our explanation makes perfect physical sense, which is obviously supported by existing experimental evidence of growing trigonal lattice structures as well as lacking the evidence of growing hexagonal lattice structures. We did not cited the above two specific references on the topic, because we already cited quite a few other similar works appeared before these two works. They all have the same hexagonal lattice and Fermi level lying in the gap, so we did not see the need to compare specifically to these two works cited by the reviewer. We would like to also point out that it would be easier to observe QSH effect in a system with Fermi level lying in the band gap, but only if the system could be made first! In other words, one would rather have a makeable system with non-ideal Fermi level than a non-makeable system with ideal Fermi level. After all, our revised system does have Fermi level lying in the gap. Anyway, as a courtesy, we added these two references along with others already cited.